# QFC: A Parallel Software Tool for Feature Construction, Based on Grammatical Evolution

**Ioannis G. Tsoulos**

Department of Informatics and Telecommunications, University of Ioannina, 47100 Arta, Greece; itsoulos@uoi.gr

**Abstract:** This paper presents and analyzes a programming tool that implements a method for classification and function regression problems. This method builds new features from existing ones with the assistance of a hybrid algorithm that makes use of artificial neural networks and grammatical evolution. The implemented software exploits modern multi-core computing units for faster execution. The method has been applied to a variety of classification and function regression problems, and an extensive comparison with other methods of computational intelligence is made.

**Keywords:** neural networks; genetic algorithms; grammatical evolution; feature construction

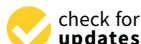



## 1. Introduction

Many problems from various research areas can be considered as classification or regression problems, such as problems from physics [1–4], chemistry [5–7], economics [8,9], pollution [10–12], medicine [13,14], etc. These problems are usually tackled by learning models such as Artificial Neural Networks [15,16], Radial Basis Function (RBF) networks [17,18], Support Vector Machines (SVM) [19], Parse-matrix evolution [20], Multilevel Block Building [21], Development of Mathematical Expressions [22], etc. A review of the methods used in classification can be found in the work of Kotsiantis et al. [23].

Learning data are usually divided into two parts: training data and test data. Learning models adjust their parameters, taking the training data as input, and are evaluated on the test data. The number of learning model parameters directly depends on the dimension of the input problem (number of features) and this means that for large problems, large amounts of memory are required to store and manage the learning models. In addition, as the number of parameters of the computational models grows, a longer time is required to adjust the parameters. Moreover, as the dimension of the data grows, more samples (patterns) are required in order to achieve high learning rates. A discussion on how the dimensionality of the input problems affects the effectiveness of neural networks is presented in [24]. A common approach to reduce the dimension of the input data is the Principal Component Analysis (PCA) technique [25–27] or the Minimum redundancy Feature Selection (MRMR) technique [28,29]. Furtheromre, Wang et al. [30] proposed an auto-encoder based dimensionality reduction method for large datasets. An overview of dimensionality reduction techniques can be found in the work of Ayesha et al. [31].

The current article describes the method and the software associated with a feature construction method based on grammatical evolution [32], which is an evolutionary process that can create programs in any programming language. The described method constructs subsets of features from the original ones using non-linear combinations of them. The method is graphically illustrated in Figure 1. Initially, the method was described in [33], and it has been utilized in a variety of cases, such as spam identification [34], fetal heart classification [35], epileptic oscillations in clinical intracranial electroencephalograms [36], classification of EEG signals [37], etc.

Feature construction methods have been thoroughly examined and analyzed in the relevant literature such as the work of Smith and Bull [38], where tree genetic programming

is used to construct artificial features. Devi uses the Simulated Annealing method [39] to identify the features that are most important for data classification. Neshatian et al. [40] construct artificial features using an entropy based fitness function for the associated genetic algorithm. Li and Yin use another evolutionary approach for feature selection using gene expression data [41]. Furthermore, Ma and Teng proposed [42] a genetic programming approach that utilizes information gain ratio (IGR) to construct artificial features.

The proposed software has been implemented in ANSI C++ utilizing the freely available library of QT from https://www.qt.io (accessed on 18 August 2022). The user should supply the training and test data of the underlying problem as well as the desired number of features that will be created. The evaluation of the constructed features can be made using a variety of machine learning models, and the user can easily extend the software to add more learning models. Moreover, the software has a variety of command line options to control the parameters of the learning models or to manage the output of the method. Finally, since the process of grammatical evolution can require a lot of execution time, parallel computation is included in the proposed software through the OpenMP programming library [43].

The proposed method differs from similar ones as it does not require any prior knowledge of the objective problem and can be applied without any change to both classification problems and regression problems. In addition, the method can discover any functional dependencies between the initial features and can drastically reduce the number of input features, significantly reducing the time required to train the subsequent machine learning model.

Related freely available software packages on feature selection and construction are also the Mlpack package [44], which implements the PCA method; the GSL software package obtained from https://www.gnu.org/software/gsl/doc/html/index.html (accessed on 18 August 2022), which also implements the PCA method, among others; the MRMR package writen in ANSI C++ by Hanchuan Peng [28,29]; etc.

The rest of this article is organized as follows: in Section 2, the grammatical evolution procedure is briefly described and the proposed method is analyzed; in Section 3, the proposed software is outlined in detail; in Section 4, a variety of experiments are conducted and presented; and finally in Section 5, some conclusions and future guidelines are discussed.

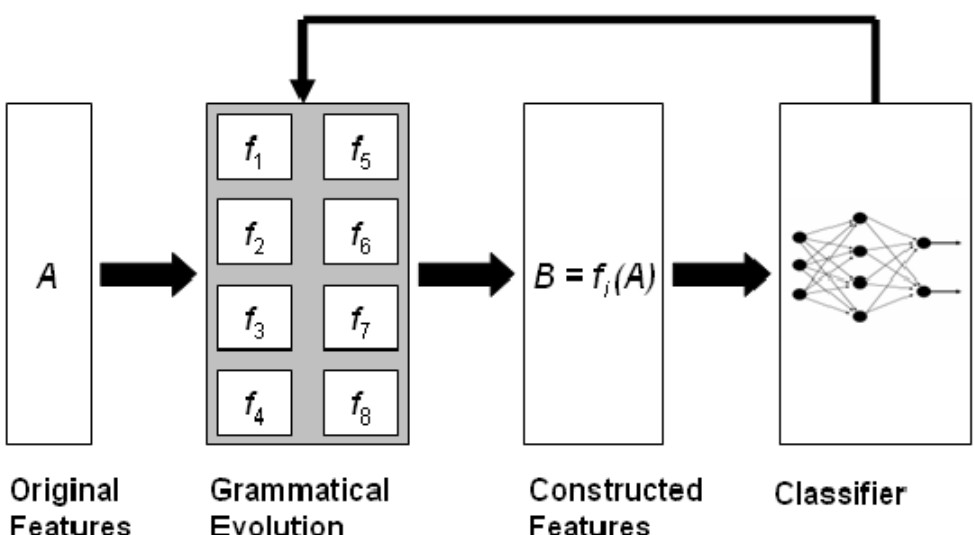

**Figure 1.** Schematic representation of the feature construction technique.

## 2. Methods

In this section, a brief overview of grammatical evolution, its applications and its advantages is given and then presented: the creation of artificial features using grammatical

evolution, the first phase of the method where features are constructed and evaluated, and the second phase of the method, where the characteristics from the first phase are evaluated.

### 2.1. Grammatical Evolution

Grammatical evolution is a biologically inspired procedure that can create artificial programs in any language. In grammatical evolution, the chromosomes enclose production rules from a BNF (Backus–Naur form) grammar [45]. These grammars are usually described as a set $G = (N, T, S, P)$, where

- $N$ is the set of non-terminal symbols.
- $T$ is the set of terminal symbols.
- $S$ is a non-terminal symbol defined as the start symbol of the grammar.
- $P$ is a set of production rules in the form $A \to a$ or $A \to aB$, $A, B \in N$, $a \in T$.

In order for grammatical evolution to work, the original grammar is expanded by enumerating all production rules. For example, consider the modified grammar of Figure 2. The symbols that are in <> are considered as non-terminal symbols. The numbers in parentheses are the production sequence numbers for each non-terminal symbol. The constant N is the original number of features for the input data. In grammatical evolution, the chromosomes are expressed as vectors of integers. Every element of each chromosome denotes a production rule from the provided BNF grammar. The algorithm starts from the start symbol of the grammar and gradually produces some program string by replacing non-terminal symbols with the right hand of the selected production rule. The selection of the rule has two steps:

- Take the next element from the chromosome and denote it as V.
- Select the next production rule according to the the scheme Rule = V mod R, where R is the number of production rules for the current non-terminal symbol.

For example, consider the chromosome

$$x = [9, 8, 6, 4, 16, 10, 17, 23, 8, 14]$$

and $N = 3$. The steps of mapping this chromosome to the valid expression $f(x) = x_2 + \cos(x_3)$ are illustrated in Table 1.

Initially, grammatical evolution was used in cases of learning functions [46,47] and solving trigonometric identities [48], but then it was also applied in other fields such as automatic composition of music [49], construction of neural networks [50,51], automatic constant creation [52], evolution of video games [53,54], energy demand estimation [55], combinatorial optimization [56], cryptography [57], etc.

A key advantage of the grammatical evolution is its easy adaptability to a wide range of problems, as long as the grammar of the problem and a fitness method are provided. No additional knowledge of the problem is required such as using derivatives. Furthermore, the method can be easily parallelized, since it is essentially a genetic algorithm of integer values. However, there are a number of disadvantages that must be taken into account when using the technique. In principle, in many cases a chromosome may not be able to produce a valid expression in the underlying grammar if its elements run out. In this case, wrapping an effect can be executed, but it is not always certain that this can again provide a valid solution. Moreover, another important issue is the initialization of the chromosomes of grammatical evolution. Usually the rules are very few in number and therefore, different numbers on the chromosomes may produce the same rules.

In the next subsection, the steps of producing artificial features from the original ones are provided and discussed.

```
S::=<expr>    (0)
<expr> ::=  (<expr> <op> <expr>)  (0)
            | <func> ( <expr> )     (1)
            |<terminal>            (2)
<op> ::=      +      (0)
            | -      (1)
            | *      (2)
            | /      (3)
<func> ::=   sin  (0)
            | cos  (1)
            |exp   (2)
            |log   (3)
<terminal>::=<xlist>                (0)
            |<digitlist>.<digitlist> (1)
<xlist>::=x1     (0)
            | x2 (1)
            .........
            | xN (N)
<digitlist>::=<digit>                (0)
            | <digit><digit>         (1)
            | <digit><digit><digit>  (2)
<digit>  ::= 0 (0)
            | 1 (1)
            | 2 (2)
            | 3 (3)
            | 4 (4)
            | 5 (5)
            | 6 (6)
            | 7 (7)
            | 8 (8)
            | 9 (9)
```

**Figure 2.** BNF grammar of the proposed method.

**Table 1.** Steps to produce a valid expression from the BNF grammar.

| String | Chromosome | Operation |
|---|---|---|
| <expr> | 9,8,6,4,16,10,17,23,8,14 | $9 \bmod 3 = 0$ |
| (<expr><op><expr>) | 8,6,4,16,10,17,23,8,14 | $8 \bmod 3 = 2$ |
| (<terminal><op><expr>) | 6,4,16,10,17,23,8,14 | $6 \bmod 2 = 0$ |
| (<xlist><op><expr>) | 4,16,10,17,23,8,14 | $4 \bmod 3 = 1$ |
| (x2<op><expr>) | 16,10,17,23,8,14 | $16 \bmod 4 = 0$ |
| (x2+<expr>) | 10,17,23,8,14 | $10 \bmod 3 = 1$ |
| (x2+<func>(<expr>)) | 17,23,8,14 | $17 \bmod 4 = 1$ |
| (x2+cos(<expr>)) | 23,8,14 | $23 \bmod 2 = 1$ |
| (x2+cos(<terminal>)) | 8,14 | $8 \bmod 2 = 0$ |
| (x2+cos(<xlist>)) | 14 | $14 \bmod 3 = 2$ |
| (x2+cos(x3)) | | |

### 2.2. The Feature Construction Procedure

The proposed technique is divided into two phases: in the first phase, new features are constructed from the old ones using grammatical evolution and in the second phase, these new features modify the control set, and a machine learning model is applied to the

new control set. The following procedure is executed in order to produce $N_f$ features from the original ones for a given chromosome $X$:

1. **Split** $X$ into $N_f$ parts.
2. **For** $i = 1 \ldots N_f$, **denote** each part as $x_i$.
3. **For** every part $x_i$, construct a feature $FT_i$ using the grammar given in Figure 2.

Every feature $FT_i$ is considered as a mapping function that transforms the original features to a new one. For example, the feature

$$FT_1 = x_1^2 + \sin((x_2)$$

is a non-linear function that maps the original feature $(x_1, x_2)$ into $FT_1$. Let $(x_1, x_2) = (2, 1)$. The mapping procedure will create the value $4 + \sin(1)$.

*2.3. The Feature Construction Step*

This is the first phase of the proposed method and it has the following steps:

1. **Initialization** step.

   (a) **Read** the train data. The train data contain $M$ patterns as pairs $(x_i, t_i)$, $i = 1 \ldots M$, where $t_i$ is the actual output for pattern $x_i$.
   (b) **Set** $N_G$, the maximum number of generations.
   (c) **Set** $N_C$, the number of chromosomes.
   (d) **Set** $p_S$, the selection rate.
   (e) **Set** $N_f$, the desired number of features.
   (f) **Set** $p_M$, the mutation rate.
   (g) **Initialize** the chromosomes of the population. Every element of each chromosome is initialized randomly in the range $[0, 255]$.
   (h) **Set** iter = 1.

2. **Genetic step.**

   (a) **For** $i = 1, \ldots, N_g$ **do**

      i. **Create**, using the procedure of Section 2.2, a set of $N_f$ for the corresponding chromosome $g_i$.
      ii. **Transform** the original train data to the new train data using the previously created features. Denote the new train set as $(x_{g_i,j}, t_j)$, $j = 1, \ldots, M$.
      iii. **Apply** a learning model $C$ (such as RBF) to the new data and **calculate** the fitness $f_i$ as

      $$f_i = \sum_{j=1}^{M} \left( C\left(x_{g_i,j}\right) - t_j \right)^2 \tag{1}$$

      iv. **Apply** the selection procedure. During selection, the chromosomes are classified according to their fitness. The best $(1 - p_s) \times N_C$ chromosomes are transferred without changes to the next generation of the population. The rest will be replaced by chromosomes that will be produced at the crossover.
      v. **Apply** the crossover procedure. During this process, $p_s \times N_c$ chromosomes will be created. Firstly, for every pair of produced offsprings, two distinct chromosomes (parents) are selected from the current population using tournament selection: First, a subset of $K > 1$ randomly selected chromosomes is created and the chromosome with the best fitness value is selected as parent. For every pair $(z, w)$ of parents, two new offsprings $\tilde{z}$ and $\tilde{w}$ are created through one point crossover as graphically shown in Figure 3.
      vi. **Apply** the mutation procedure. For every element of each chromosome, select a random number $r \in [0, 1]$ and alter the corresponding chromosome if $r \leq p_m$.

(b)　**End For**

3.　**Set** iter = iter+1.
4.　**If** iter $\leq N_G$, goto **Genetic** Step, else **terminate** and obtain $g^*$ as the best chromosome in the population.

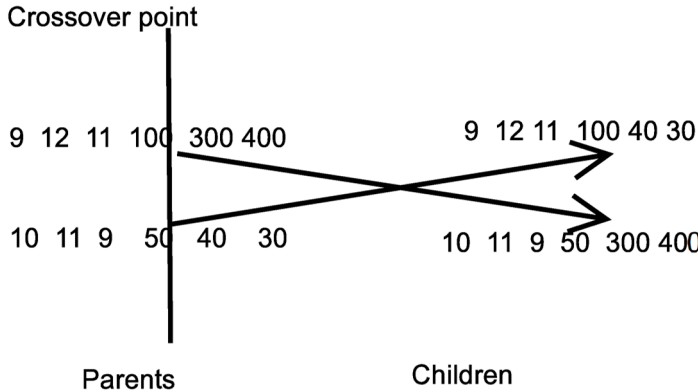

**Figure 3.** Example of one-point crossover.

*2.4. The Feature Evaluation Step*

During the feature evaluation step, the following steps are executed:

1.　**Denote** as $T = (x_i, y_i)$, $i = 1, \ldots, K$ the original test set.
2.　**Obtain** the best chromosome $g^*$ of the feature construction step.
3.　**Construct** $N_f$ features for $g^*$ using the procedure of Section 2.2.
4.　**Transform** $T$ into $T' = (x_{g^*,i}, y_i)$, $i = 1, \ldots, K$ using the previously constructed features.
5.　**Apply** a learning model such as RBF or a neural network to $T'$ and obtain the test error.

**3. The Software**

The proposed method has been fully implemented in ANSI C++ and is freely available from the internet. This section also acts as a small manual for this software. It starts with the software installation instructions, then all the operating parameters of the software are presented and the use of the software is demonstrated through an analytical example.

*3.1. Installation Procedure*

The software is entirely written in ANSI C++ using the freely available QT programming library. The library can be downloaded from https://www.qt.io (accessed on 18 August 2022) As expected, the software can be installed on the majority of operating systems, even on mobile devices (Android, iOS, etc.). The program is freely available from https://github.com/itsoulos/QFc (accessed on 18 August 2022), and the user should issue the following commands under most UNIX systems to compile the project:

1.　Download QFc-master.zip from the above url
2.　gunzip QFc-master.zip
3.　cd QFc
4.　qmake.
5.　make

The final outcome of the previous steps is a command line program called *qfc*. This program has a series of command line parameters that is illustrated subsequently.

*3.2. The Program qfc*

The program *qfc* has a variety of command line options. All options are in the form $--$key = value, where key is the name of the option and value is the actual option value. The main options of the program are:

1.  $--$trainFile = filename, where filename is the full path to data containing the input train set. The file must be in the format of Figure 4. The integer number D denotes the number of features of the dataset, and M represents the number of patterns. In every subsequent line of the file, there should be the input pattern, and the final column is the real output (category) for the corresponding pattern.

2.  $--$testFile = filename, where filename is the full path to data containing the input test set. The format of this file should be the same as the train data. The user should at least provide the train and test set in order to execute the program.

3.  $--$features = n, set as the n the number of features that will be constructed by the method. The default value is 1.

4.  $--$randomSeed = r, set as r the random seed generator. The default value for this parameter is 1, and the drand48() random generator of c++ language was used.

5.  $--$featureCreateModel = model, the string parameter model sets the name of the used feature construction model. The default value is "rbf", and accepted values are:

    (a)  **copy**. With this value, no feature construction is done and the original dataset is used for training by the model specified by the option $--$featureEvaluate Model.

    (b)  **rbf**. This value is used to utilize a Radial Basis Function neural network for the evaluation of the constructed features.

    (c)  **neural**. This value is used to use a neural network for the evaluation of the constructed features.

    (d)  **knn**. A simple K-nearest neighbor (KNN) method is used [58].

    (e)  **osamaRbf**. A simple RBF implementation as downloaded from https://github.c om/osama-afifi/RBF-Radial-Basis-Function-Network (accessed on 18 August 2022).

6.  $--$featureEvaluateModel = model, the string parameter model sets the name of the used model for the evaluation of the constructed features. The default value is "neural", but other accepted values are: rbf, knn, osamaRbf, nnc. The value nnc refers to the Neural Network Construction model as proposed by Tsoulos et al. [59].

7.  $--$threads=t, the number of OpenMP threads used. The default value is 1.

8.  $--$neural_trainingMethod = m. This is the option that defines the method used for neural network training. The default value for m is "bfgs", and accepted values are

    (a)  **bfgs**. This value sets as a training method a BFGS variant of Powell [60].

    (b)  **lbfgs**. This value sets as a training method the limited memory BFGS [61,62].

    (c)  **genetic**. With this value, a simple genetic algorithm [63,64] is used to train the neural network.

9.  $--$neural_weights = n, the weights used in neural networks. The default value is 1.

10. $--$knn_weights = n, the weights (neighbors) used in the knn model. The default value is 1.

11. $--$rbf_weights = n, the weights used in the rbf model.

12. $--$ge_chromosomes = n, the number of chromosomes in the grammatical evolution procedure. The default value is 500.

13. $--$ge_maxGenerations = n, the maximum number of generations for the grammatical evolution procedure. The default value is 200.

14. $--$ge_selectionRate = f, the selection rate used in the grammatical evolution procedure. The default value is 0.10 (10%).

15. $--$ge_mutationRate = f, the mutation rate used in the grammatical evolution procedure. The default value is 0.05 (5%).

16. $--$ge_length = n, the length of chromosomes in the grammatical evolution procedure. The default value is $40 \times d$, where $d$ is the number of features that will be created.

17. $--$genetic_chromosomes = n, the number of chromosomes used in the genetic algorithm for neural network training. The default value is 500.

18. $--$genetic_maxGenerations = n, the maximum number of generations for the genetic algorithm used for neural network training. The default value is 200.

19.  $--$genetic_selectionRate = f, the selection rate used in the genetic algorithm of neural network training. The default value is 0.1 (10%).
20.  $--$genetic_mutationRate = f, the mutation rate used in the genetic algorithm of neural network training. The default value is 0.05 (5%).
21.  $--$bfgs_iterations = n, the maximum number of iterations for the BFGS method. The default value is 2001.
22.  $--$export_train_file = f. The value f specifies the file where the training set will be exported after new features are constructed. The new file will have the same format and the same number of templates as the original, but the dimension will be changed to the one defined with the parameter $--$features.
23.  $--$export_test_file = f, the value f specifies the file where the test set will be exported after new features are constructed. The new file will have the same format and the same number of templates as the original, but the dimension will be changed to the one defined with the parameter $--$features.
24.  $--$export_cpp_file = f, where f is the output of the constructed features in the C++ programming language. As an example, consider the file outlined in Figure 5. The function fcMap() is a function with two array arguments:

    (a)  The argument inx denotes an input pattern with the original dimension. For the case of BL dataset, the original dimension is 7.
    (b)  The argument outx stands for the features created by the algorithm. In this case, outx[0] is the first feature and outx[1] is the second feature.

25.  $--$help, prints a help screen and terminates the program.

$$
\begin{array}{ccccc}
D & & & & \\
M & & & & \\
x_{11} & x_{12} & \dots & x_{1D} & y_1 \\
x_{21} & x_{22} & \dots & x_{2D} & y_2 \\
\vdots & \vdots & \vdots & \vdots & \vdots \\
x_{M1} & x_{M2} & \dots & x_{MD} & y_M
\end{array}
$$

**Figure 4.** Example of input file for regression/classification.

```
#include <math.h>
void fcMap(double *inx,double *outx)
{
        double x1=inx[0];
        double x2=inx[1];
        double x3=inx[2];
        double x4=inx[3];
        double x5=inx[4];
        double x6=inx[5];
        double x7=inx[6];
        outx[0]=exp((5.4*x1/(5.3*x4*(38.599*x1*(x7*cos(((-55.68)/5.4)*x1))))));
        outx[1]=((-88.007)*x3/(9.998*x7+cos(((-8.81)*x5*sqrt((783.138*x2))))));
}
```

**Figure 5.** An example output file for the BL dataset.

### 3.3. Example Run

As an example run, consider the wdbc dataset located in the *examples* folder of the distribution. The following command:

```
./QFc --trainFile = examples/wdbc.train --testFile =
examples/wdbc.test
      --features = 2 --ge_maxGenerations=5
      --featureEvaluatedModel=rbf --threads = 8
```

will produce the following output

```
Iteration:  1  Best Fitness:  -52.1562
Best program:  f1(x)=(447.63*x7-x14+x15)
Iteration:  2  Best Fitness:  -52.1562
Best program:  f1(x)=(447.63*x7-x14+x15)
Iteration:  3  Best Fitness:  -51.8369
Best program:  f1(x)=(417.63*x7-x14+x15)
Iteration:  4  Best Fitness:  -51.7176
Best program:  f1(x)=(417.63*x7-x14+(48.07/(-486.503))*x14+exp(55.884*x15))
Iteration:  5  Best Fitness:  -51.5149
Best program:   f1(x)=(417.63*x7-x14+x15+x16+log(((-9.863)/5.9)*x30+exp(sin(x5+(417.63/07.54)*x8+7.494*x29))))
AVERAGES(TRAIN,TEST,CLASS):      57.241174      60.60288      29.473684%
```

## 4. Experiments

A series of experiments were performed in order to evaluate the reliability and accuracy of the proposed methodology. In these experiments, the accuracy of the proposed methodology against other techniques, the running time of the experiments, and the sensitivity of the experimental results to various critical parameters, such as the number of features or the maximum number of generations of the genetic algorithm, were measured. All the experiments were conducted 30 times with different seeds for the random number generator each time and averages were taken. All the experiments were conducted on an AMD Ryzen 5950X equipped with 128GB of RAM. The operating system used was OpenSUSE Linux, and all the programs were compiled using the GNU C++ compiler. In all the experiments, the parameter −−featureCreateModel had the value rbf, since it is the fastest model that could be used and it has high learning rates. For classification problems, the average classification error on the test set is shown and, for regression datasets, the average mean squared error on the test set is displayed. In all cases, 10-fold cross-validation was used, and the number of parameters for neural networks and for RBF networks was set to 10. The evaluation of the features constructed by grammatical evolution was made using the Function Parser library [65].

### 4.1. Experimental Datasets

The validity of the method was tested on a series of well-known datasets from the relevant literature. The main repositories for the testing were:

1. The UCI Machine Learning Repository http://www.ics.uci.edu/~mlearn/MLRepository.html (accessed on 18 August 2022)
2. The Keel repository https://sci2s.ugr.es/keel/ (accessed on 18 August 2022)
3. The Statlib repository http://lib.stat.cmu.edu/datasets/ accessed on 18 August 2022).

The classification datasets are:

1. **Australian** dataset [66], a dataset related to credit card applications.
2. **Alcohol** dataset, a dataset about Alcohol consumption [67].
3. **Balance** dataset [68], which is used to predict psychological states.
4. **Cleveland** dataset, a dataset used to detect heart disease and used in various papers [69,70].
5. **Dermatology** dataset [71], which is used for differential diagnosis of erythemato-squamous diseases.
6. **Glass** dataset. This dataset contains glass component analysis for glass pieces that belong to six classes.
7. **Hayes Roth** dataset [72]. This dataset contains **5** numeric-valued attributes and 132 patterns.
8. **Heart** dataset [73], used to detect heart disease.
9. **HouseVotes** dataset [74], which is about votes in the U.S. House of Representatives Congressmen.
10. **Liverdisorder** dataset [75], used to detect liver disorders in peoples using blood analysis.
11. **Ionosphere** dataset, a meteorological dataset used in various research papers [76,77].
12. **Mammographic** dataset [78]. This dataset can be used to identify the severity (benign or malignant) of a mammographic mass lesion from BI-RADS attributes and the patient's age. It contains 830 patterns of 5 features each.

13. **PageBlocks** dataset. The dataset contains blocks of the page layout of a document that has been detected by a segmentation process. It has 5473 patterns with 10 features each.
14. **Parkinsons** dataset, ref. [79], which is created using a range of biomedical voice measurements from 31 people, 23 with Parkinson's disease (PD). The dataset has 22 features.
15. **Pima** dataset [80], used to detect the presence of diabetes.
16. **PopFailures** dataset [81], used in meteorology.
17. **Regions2** dataset. It is created from liver biopsy images of patients with hepatitis C [82]. From each region in the acquired images, 18 shape-based and color-based features were extracted, while it was also annotated from medical experts. The resulting dataset includes 600 samples belonging to 6 classes.
18. **Saheart** dataset [83], used to detect heart disease.
19. **Segment** dataset [84]. This database contains patterns from a database of seven outdoor images (classes).
20. **Sonar** dataset [85]. The task here is to discriminate between sonar signals bounced off a metal cylinder and those bounced off a roughly cylindrical rock.
21. **Spiral** dataset, which is an artificial dataset with two classes. The features in the first class are constructed as: $x_1 = 0.5t \cos(0.08t)$, $x_2 = 0.5t \cos\left(0.08t + \frac{\pi}{2}\right)$ and for the second class the used equations are : $x_1 = 0.5t \cos(0.08t + \pi)$, $x_2 = 0.5t \cos\left(0.08t + \frac{3\pi}{2}\right)$.
22. **Wine** dataset, which is related to chemical analysis of wines [86,87].
23. **Wdbc** dataset [88], which contains data for breast tumors.
24. **EEG** dataset. As a real-world example, an EEG dataset described in [89,90] is used here. The dataset consists of five sets (denoted as Z, O, N, F, and S), each containing 100 single-channel EEG segments and each having a 23.6 sec duration. Sets Z and O have been taken from surface EEG recordings of five healthy volunteers with eye open and closed, respectively. Signals in two sets have been measured in seizure-free intervals from five patients in the epileptogenic zone (F) and from the hippocampal formation of the opposite hemisphere of the brain (N). Set S contains seizure activity, selected from all recording sites exhibiting ictal activity. Sets Z and O have been recorded extracranially, whereas sets N, F, and S have been recorded intracranially.
25. **Zoo** dataset [91], where the task is to classify animals in seven predefined classes.

The regression datasets used are the following:

1. **Abalone** dataset [92]. This dataset can be used to obtain a model to predict the age of abalone from physical measurements.
2. **Airfoil** dataset, which is used by NASA for a series of aerodynamic and acoustic tests [93].
3. **Baseball** dataset, a dataset to predict the salary of baseball players.
4. **BK** dataset, used to estimate the points scored per minute in a basketball game.
5. **BL** dataset, which is related to the effects of machine adjustments on the time to count bolts.
6. **Concrete** dataset. This dataset is taken from civil engineering [94].
7. **Dee** dataset, used to predict the daily average price of the electricity energy in Spain.
8. **Diabetes** dataset, a medical dataset.
9. **FA** dataset, which contains percentage of body fat and ten body circumference measurements. The goal is to fit body fat to the other measurements.
10. **Housing** dataset. This dataset was taken from the StatLib library and it is described in [95].
11. **MB** dataset. This dataset is available from Smoothing Methods in Statistics [96], and it includes 61 patterns.
12. **MORTGAGE** dataset, which contains economic data information from the USA.
13. **NT** dataset [97], which is related to the body temperature measurements.
14. **PY** dataset [98], used to learn Quantitative Structure Activity Relationships (QSARs).
15. **Quake** dataset, used to estimate the strength of an earthquake.

16. **Treasure** dataset, which contains economic data information from the USA from 1 April 1980 to 2 April 2000 on a weekly basis.
17. **Wankara** dataset, which contains weather information.

### 4.2. Experimental Results

The parameters for the used methods are listed in Table 2. In all tables, an additional row was added at the end showing the average classification or regression error for all datasets, and it is denoted by the name AVERAGE. The columns of all tables have the following meaning:

1. The column RBF stands for the results from an RBF network with $H$ Gaussian units.
2. The column MLP stands for the results of a neural network with $H$ sigmoidal nodes trained by a genetic algorithm. The parameters of this genetic algorithm are listed in Table 2.
3. The column FCRBF represents the results of the proposed method, when an RBF network with $H$ Gaussian units was used as the evaluation model.
4. The column FCMLP represents the results of the proposed method, when a neural network trained by a genetic algorithm was used as the evaluation model. The parameters of this genetic algorithm are listed in Table 2.
5. The column FCNNC stands for the results of the proposed method, when the neural network construction model (nnc) was utilized as the evaluation model.
6. The column MRMR stands for the Minimum Redundancy Maximum Relevance Feature Selection method with two selected features. The features selected by MRMR are evaluated using an artificial neural network trained by a genetic algorithm using the parameters of Table 2.
7. The Principal Component Analysis (PCA) method, as implemented in Mlpack software [44], was used to construct two features. The features constructed by PCA are evaluated using an artificial neural network trained by a genetic algorithm using the parameters of Table 2.

**Table 2.** Experimental parameters.

| PARAMETER | MEANING | VALUE |
|:---:|:---:|:---:|
| $H$ | Neural weights | 10 |
| $N_C$ | Chromosomes | 500 |
| $N_F$ | Features | 2 |
| $p_S$ | Selection rate | 0.10 |
| $p_M$ | Mutation rate | 0.05 |
| $N_G$ | Generations | 200 |

The experimental results for the classification datasets are listed in Table 3, and for regression datasets in Table 4. Furtheromre, a comparison against MRMR and PCA is performed in Tables 5 and 6, respectively.

Summarizing the conclusions of the experiments, one can say that the proposed method obviously outperforms the other techniques in most cases, especially in the case of regression datasets. In the case of data classification, there is a gain of the order of 30%, and in the case of regression data, the gain from the application of the proposed technique exceeds 50%. The gain in many cases from the application of the proposed technique can even reach 90%. In addition, the MRMR method seems to be superior in most cases to the PCA, possibly pointing the way for a future research on the combination of grammatical evolution and MRMR. Furthermore, among the three cases of models used to evaluate the constructed features (FCRBF, FCMLP, FCNNC), there does not seem to be any clear superiority of any of the three. However, we would say that the nnc method slightly outperforms the simple genetic algorithm.

**Table 3.** Experimental results between the method and other techniques for the classification datasets.

| DATASET | RBF | MLP | FCRBF | FCMLP | FCNNC |
|---|---|---|---|---|---|
| ALCOHOL | 49.19% | 47.49% | 35.56% | 26.57% | 28.58% |
| AUSTRALIAN | 34.89% | 32.21% | 15.37% | 14.31% | 14.24% |
| BALANCE | 33.42% | 8.97% | 14.39% | 1.42% | 1.52% |
| DERMATOLOGY | 62.34% | 30.58% | 22.42% | 15.06% | 15.42% |
| GLASS | 50.16% | 60.25% | 49.81% | 55.94% | 52.62% |
| HAYES ROTH | 64.36% | 56.18% | 34.59% | 29.58% | 30.67% |
| HEART | 31.20% | 28.34% | 18.61% | 15.67% | 17.21% |
| HOUSEVOTES | 5.99% | 6.62% | 7.15% | 5.22% | 3.96% |
| IONOSPHERE | 16.22% | 15.14% | 9.83% | 9.48% | 9.92% |
| LIVERDISORDER | 30.84% | 31.11% | 30.77% | 31.98% | 30.24% |
| MAMMOGRAPHIC | 21.38% | 19.88% | 16.68% | 17.92% | 16.75% |
| PAGEBLOCKS | 10.09% | 8.06% | 9.24% | 5.58% | 5.85% |
| PARKINSONS | 17.41% | 18.05% | 8.48% | 10.82% | 12.53% |
| PIMA | 25.75% | 32.19% | 24.07% | 30.02% | 25.01% |
| POPFAILURES | 7.04% | 5.94% | 4.94% | 4.94% | 4.43% |
| REGIONS2 | 37.49% | 29.39% | 25.49% | 27.52% | 24.40% |
| SAHEART | 32.19% | 34.86% | 29.10% | 27.91% | 27.17% |
| SEGMENT | 59.69% | 57.72% | 39.35% | 49.52% | 46.14% |
| SONAR | 27.85% | 26.97% | 24.35% | 25.38% | 23.68% |
| SPIRAL | 44.87% | 45.77% | 34.34% | 45.53% | 42.69% |
| TAE | 60.07% | 56.22% | 50.95% | 56.87% | 55.67% |
| WDBC | 7.27% | 8.56% | 3.39% | 4.36% | 4.51% |
| WINE | 31.41% | 19.20% | 7.61% | 11.08% | 11.61% |
| Z_F_S | 13.16% | 10.73% | 5.48% | 6.72% | 6.63% |
| ZO_NF_S | 9.02% | 8.41% | 4.08% | 4.25% | 4.34% |
| ZONF_S | 4.03% | 2.60% | 1.89% | 4.62% | 3.18% |
| Z_O_N_F_S | 48.71% | 65.45% | 39.29% | 40.93% | 41.19% |
| ZOO | 21.77% | 16.67% | 26.07% | 13.30% | 10.33% |
| AVERAGE | 31.89% | 28.80% | 22.11% | 22.02% | 21.22% |

In addition, one more experiment was done in order to establish the impact of the number of features on the accuracy of the proposed method. In this case, the RBF (FCRBF) network was used as the feature evaluator, and the number of generated features was in the interval [1...4]. The average classification error and the average regression error for all datasets are shown in Table 7. From the experimental results, the robustness of the proposed methodology is clearly visible, as one or two features seem to be enough to achieve high learning rates for the experimental data used.

Furthermore, one more experiment was conducted to determine the effect of the maximum number of generations on the accuracy of the proposed method. Again, as an evaluator model, the RBF was used. The number of generations was varied from 50 to 400, and the average classification error and average regression error for all datasets were measured. The results for this experiment are presented in Table 8. Once again, the dynamics of the proposed method appear as a few generations are enough to achieve high learning rates.

**Table 4.** Experiments for regression datasets.

| DATASET | RBF | MLP | FCRBF | FCMLP | FCNNC |
|---------|-----|-----|-------|-------|-------|
| ABALONE | 7.32 | 7.17 | 4.46 | 4.18 | 4.39 |
| AIRFOIL | 0.05 | 0.003 | 0.002 | 0.001 | 0.001 |
| BASEBALL | 78.89 | 103.60 | 48.04 | 52.50 | 51.40 |
| BK | 0.02 | 0.03 | 0.02 | 0.02 | 0.02 |
| BL | 0.01 | 5.74 | 0.04 | 0.001 | 0.01 |
| CONCRETE | 0.01 | 0.01 | 0.006 | 0.004 | 0.005 |
| DEE | 0.17 | 1.01 | 0.18 | 0.40 | 0.38 |
| DIABETES | 0.49 | 19.86 | 1.49 | 0.58 | 0.61 |
| HOUSING | 57.68 | 43.26 | 12.78 | 28.47 | 17.47 |
| FA | 0.01 | 1.95 | 0.01 | 0.02 | 0.01 |
| MB | 1.91 | 3.39 | 0.48 | 0.12 | 0.06 |
| MORTGAGE | 1.45 | 2.41 | 0.66 | 1.37 | 0.22 |
| NT | 8.15 | 0.05 | 0.25 | 0.007 | 0.02 |
| PY | 0.02 | 105.41 | 0.17 | 0.03 | 0.03 |
| QUAKE | 0.07 | 0.04 | 0.06 | 0.02 | 0.04 |
| TREASURY | 2.02 | 2.93 | 0.29 | 1.41 | 0.11 |
| WANKARA | 0.001 | 0.012 | 0.0004 | 0.0002 | 0.0002 |
| AVERAGE | 9.31 | 17.46 | 4.06 | 5.24 | 4.4 |

**Table 5.** Comparison against MRMR and PCA for the classification datasets.

| DATASET | MRMR | PCA | FCRBF | FCMLP | FCNNC |
|---------|------|-----|-------|-------|-------|
| ALCOHOL | 56.75% | 70.29% | 35.56% | 26.57% | 28.58% |
| AUSTRALIAN | 32.92% | 49.97% | 15.37% | 14.31% | 14.24% |
| BALANCE | 56.80% | 56.48% | 14.39% | 1.42% | 1.52% |
| DERMATOLOGY | 68.54% | 62.11% | 22.42% | 15.06% | 15.42% |
| GLASS | 58.35% | 50.16% | 49.81% | 55.94% | 52.62% |
| HAYES ROTH | 61.21% | 61.13% | 34.59% | 29.58% | 30.67% |
| HEART | 38.04% | 35.84% | 18.61% | 15.67% | 17.21% |
| HOUSEVOTES | 3.05% | 10.80% | 7.15% | 5.22% | 3.96% |
| IONOSPHERE | 12.93% | 21.22% | 9.83% | 9.48% | 9.92% |
| LIVERDISORDER | 40.32% | 45.01% | 30.77% | 31.98% | 30.24% |
| MAMMOGRAPHIC | 16.84% | 17.23% | 16.68% | 17.92% | 16.75% |
| PAGEBLOCKS | 14.91% | 13.05% | 9.24% | 5.58% | 5.85% |
| PARKINSONS | 17.16% | 16.96% | 8.48% | 10.82% | 12.53% |
| PIMA | 26.29% | 39.43% | 24.07% | 30.02% | 25.01% |
| POPFAILURES | 7.04% | 31.42% | 4.94% | 4.94% | 4.43% |
| REGIONS2 | 33.31% | 32.50% | 25.49% | 27.52% | 24.40% |
| SAHEART | 28.78% | 36.96% | 29.10% | 27.91% | 27.17% |
| SEGMENT | 45.72% | 70.29% | 39.35% | 49.52% | 46.14% |
| SONAR | 43.92% | 49.97% | 24.35% | 25.38% | 23.68% |

**Table 5.** *Cont.*

| DATASET | MRMR | PCA | FCRBF | FCMLP | FCNNC |
|---------|------|-----|-------|-------|-------|
| SPIRAL | 44.87% | 45.94% | 34.34% | 45.53% | 42.69% |
| TAE | 61.00% | 64.80% | 50.95% | 56.87% | 55.67% |
| WDBC | 12.91% | 10.28% | 3.39% | 4.36% | 4.51% |
| WINE | 30.73% | 30.39% | 7.61% | 11.08% | 11.61% |
| Z_F_S | 32.71% | 44.81% | 5.48% | 6.72% | 6.63% |
| ZO_NF_S | 33.79% | 40.02% | 4.08% | 4.25% | 4.34% |
| ZONF_S | 10.31% | 12.63% | 1.89% | 4.62% | 3.18% |
| Z_O_N_F_S | 43.04% | 56.45% | 39.29% | 40.93% | 41.19% |
| ZOO | 19.03% | 11.50% | 26.07% | 13.30% | 10.33% |
| AVERAGE | 33.97% | 38.84% | 22.11% | 22.02% | 21.22% |

**Table 6.** Comparison against MRMR and PCA for the regression datasets.

| DATASET | MRMR | PCA | FCRBF | FCMLP | FCNNC |
|---------|------|-----|-------|-------|-------|
| ABALONE | 6.48 | 6.70 | 4.46 | 4.18 | 4.39 |
| AIRFOIL | 0.003 | 0.019 | 0.002 | 0.001 | 0.001 |
| BASEBALL | 100.21 | 101.87 | 48.04 | 52.50 | 51.40 |
| BK | 0.03 | 0.17 | 0.02 | 0.02 | 0.02 |
| BL | 0.15 | 0.19 | 0.04 | 0.001 | 0.01 |
| CONCRETE | 0.025 | 0.273 | 0.006 | 0.004 | 0.005 |
| DEE | 0.40 | 0.55 | 0.18 | 0.40 | 0.38 |
| DIABETES | 19.86 | 27.36 | 1.49 | 0.58 | 0.61 |
| HOUSING | 67.97 | 119.08 | 12.78 | 28.47 | 17.47 |
| FA | 0.02 | 0.08 | 0.01 | 0.02 | 0.01 |
| MB | 3.39 | 4.33 | 0.48 | 0.12 | 0.06 |
| MORTGAGE | 0.17 | 2.41 | 0.66 | 1.37 | 0.22 |
| NT | 0.05 | 1.57 | 0.25 | 0.007 | 0.02 |
| PY | 1.56 | 0.30 | 0.17 | 0.03 | 0.03 |
| QUAKE | 1.44 | 1.80 | 0.06 | 0.02 | 0.04 |
| TREASURY | 0.12 | 3.13 | 0.29 | 1.41 | 0.11 |
| WANKARA | 0.002 | 0.90 | 0.0004 | 0.0002 | 0.0002 |
| AVERAGE | 11.88 | 15.93 | 4.06 | 5.24 | 4.4 |

**Table 7.** Average errors regarding the number of features.

| FEATURES | AVERAGE CLASS | AVERAGE REGRESSION |
|----------|---------------|--------------------|
| 1 | 23.02% | 4.73 |
| 2 | 22.11% | 4.06 |
| 3 | 21.03% | 4.23 |
| 4 | 22.16% | 4.28 |

**Table 8.** Average error depending on the number of generations.

| GENERATIONS | AVERAGE CLASS | AVERAGE REGRESSION |
|:-----------:|:-------------:|:------------------:|
| 50 | 22.38% | 5.61 |
| 200 | 22.11% | 4.06 |
| 400 | 21.46% | 4.21 |

## 5. Conclusions

A feature construction method and the accompanying software were analyzed in detail in this paper. The software is developed in ANSI C++ and is freely available on the internet. The proposed technique constructs technical features from the existing ones by exploiting the possible functional dependencies between the features, but also the possibility that some of the original features do not contribute anything to the learning. The method does not require any prior knowledge of the objective problem and can be applied without any change to both regression and classification problems. The method is divided into two phases: in the first phase, a genetic algorithm using grammatical evolution is used to construct new features from the original ones. These features are evaluated for their accuracy with some machine learning model. However, this process can be very time consuming and as a consequence, fast learning models or parallel algorithms should be used. Radial basis networks were used in the first phase of the method during the experiments, which are known to have short training times, but other models could be used in their place. In the second phase of the method after the features are generated, a machine learning method is applied to them and the error on the test set is evaluated.

The user can choose between several learning models and can customize the course of the technique through a series of command line parameters. From the extensive execution of experiments and the comparison with other learning methods, the superiority of the proposed technique and its ability to achieve high learning rates even with a limited number of constructed features or a maximum number of iterations emerge. These results combined with the ability of the method to run on multiple threads through the OpenMP library make it ideal for learning large sets of data in a satisfactory execution time.

The method can be extended in several ways, such as:

1. Incorporation of advanced stopping rules.
2. Usage of more advanced learning models such as SVM.
3. Addition of more input formats such as the ARFF format or CSV format.
4. Incorporation of the MPI library [99] for a large network of computers.

**Funding:** This research received no external funding.

**Institutional Review Board Statement:** Not applicable.

**Informed Consent Statement:** Not Applicable.

**Data Availability Statement:** Not applicable.

**Acknowledgments:** The experiments of this research work were performed at the high-performance computing system established at Knowledge and Intelligent Computing Laboratory, Dept of Informatics and Telecommunications, University of Ioannina, acquired with the project "Educational Laboratory equipment of TEI of Epirus" with MIS 5007094 funded by the Operational Programme "Epirus" 2014–2020, by ERDF and national finds.

**Conflicts of Interest:** The author declares no conflict of interest.

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
