# Peer review of "QFC: A Parallel Software Tool for Feature Construction, Based on Grammatical Evolution"

_algorithms, doi:10.3390/a15080295_

Round 1

Reviewer 1 Report

In the manuscript, a programming tool, based on Grammatical Evolution is presented, implements a method for classification and function regression problems. The method builds new features from existing ones with the assistance of a hybrid algorithm that makes use of artificial neural networks and Grammatical Evolution. The method has been applied to a variety of classification and function regression problems and an extensive comparison with other methods.

However, I do not think the manuscript is good for publishing as a research paper. My main concerns are as follows:

- The manuscript did not provide a clear understanding/definition of grammatical evolution. The scope of this paper is not clear.

- In the introduction or section 2, the author first should give a comprehensive review of the history of the development of Grammatical Evolution, then give a classification of the existing methods based on Grammatical Evolution, and finally emphasize the specificity and superiority of the Grammatical Evolution methods.

- The manuscript did not provide a deep discussion on feature construction.

- Presenting the basic steps of Grammatical Evolution cannot be treated as a contribution of the study.

- The differences between evolutionary programming algorithms should be highlighted from different perspectives.

- The authors should provide a discussion section that provides advantages/disadvantages of Grammatical Evolution, future trends, and the overall evaluation of the method.

- The structure of section 3 is unclear, similarly, the readability of the structure of section 2 is poor.

- The conclusion of this paper does not provide valuable information. The conclusion is very brief and simply summarizes the paper, without adding any value to the field.

- The manuscript described only a partial collection of Grammatical Evolution methods. For example, notable progress including Genetic Programming (GP), other Automatic Programming (AP) methods, Parse-Matrix Evolution (PME), Multilevel Block Building (MBB), Development of Mathematical Expressions (DoME) were neglected in the context.

- In addition to all these, the results obtained are not at competitive with the results in the literature.

Author Response

1. COMMENT

In the introduction or section 2, the author first should give a comprehensive review of the history of the development of Grammatical Evolution, then give a classification of the existing methods based on Grammatical Evolution, and finally emphasize the specificity and superiority of the Grammatical Evolution methods.

RESPONSE

1) We have added a short description at the beginning of section 2 with the contents of this section.

2) The Grammatical Evolution procedure is fully described in subsection 2.1

3) The following text has been added in subsection 2.1

Initially, Grammatical Evolution was used in cases of learning functions [46, 47] and solving trigonometric identities [48] but then it was also applied in other fields such as automatic composition of music [49], construction of neural networks [50, 51], automatic constant creation [52], evolution of video games [53, 54], energy demand estimation [55], combinatorial optimization [56], cryptography [57] etc.

A key advantage of the Grammatical Evolution is its easy adaptability to a wide range of problems, as long as the grammar of the problem and a fitness method are provided. No additional knowledge of the problem is required such as using derivatives. Also, the method can be easily parallelized, since it is essentially a genetic algorithm of integer values. However, there are a number of disadvantages that must be taken into account when using the technique. In principle in many cases a chromosome may not be able to produce a valid expression in the underlying grammar if its elements run out. In this case, wrapping an effect can be executed, but it is not always certain that this can again provide a valid solution. Also another important issue is the initialization of the chromosomes of Grammatical evolution. Usually the rules are very few in number and therefore different numbers on the chromosomes may produce the same rules.In the next subsection the steps of producing artificial features from the original ones are provided and discussed.”.

2. COMMENT

The manuscript did not provide a deep discussion on feature construction.

RESPONSE

The following text has been added in the Introduction section of the revised manuscript:

Feature constuction methods have been thoroughly examined and analyzed in the relevant literature such as the work of Smith and Bull [genfc1], where tree genetic programming is used to construct artificial features. Devi uses the Simulated Annealing method[genfc_anneal] to identify the features that are most important for data classification. Neshatian et al [genfc2] constructs artificial features using an entropy based fitness function for the associated genetic algorithm. Li and Yin uses another evolutionary approach for feature selection using gene expression data [genfc3]. Furthermore, Ma and Teng proposed [genfc4] a genetic programming approach that utilizes information gain ratio (IGR) to construct artificial features.

3. COMMENT

The differences between evolutionary programming algorithms should be highlighted from different perspectives.

RESPONSE

The following text has been added in the Introduction of the revised manuscript:

The proposed method differs from similar ones as it does not require any prior knowledge of the objective problem and can be applied without any change to both classification problems and regression problems. In addition, the method can discover any functional dependencies between the initial features and can drastically reduce the number of input features, significantly reducing the time required to train the subsequent machine learning model.

4. COMMENT

The structure of section 3 is unclear.

RESPONSE

1) We have added the following text at the beginning of Section 3:

The proposed method has been fully implemented in ANSI C++ and is freely available from the internet. This section also acts as a small manual for this software. It starts with the software installation instructions, then all the operating parameters of the software are presented and the use of the software is demonstrated through an analytical example.”

2) The subsection 3.2 has been merged to subsection 3.3

5. COMMENT

The conclusion of this paper does not provide valuable information. The conclusion is very brief and simply summarizes the paper, without adding any value to the field.

RESPONSE

The following text has been added in the revised version of the manuscript:

A feature construction method and the accompanying software were analyzed in detail in this paper. The software is developed in ANSI C++ and is freely available on the internet. The proposed technique constructs technical features from the existing ones by exploiting the possible functional dependencies between the features but also the possibility that some of the original features do not contribute anything to the learning. The method does not require any prior knowledge of the objective problem and can be applied without any change to both regression and classification problems. The method is divided into two phases: in the first phase a genetic algorithm using Grammatical Evolution is used to construct new features from the original ones. These features are evaluated for their accuracy with some machine learning model. However, this process can be very time consuming and as a consequence fast learning models or parallel algorithms should be used. Radial basis networks were used in the first phase of the method during the experiments, which are known to have short training times, but other models could be used in their place. In the second phase of the method after the features are generated, a machine learning method is applied to them and the error on the test set is evaluated.”

6. COMMENT

The manuscript described only a partial collection of Grammatical Evolution methods. For example, notable progress including Genetic Programming (GP), other Automatic Programming (AP) methods, Parse-Matrix Evolution (PME), Multilevel Block Building (MBB), Development of Mathematical Expressions (DoME) were neglected in the context.

RESPONSE

Appropriate references have been added in the revised version of the manuscript.

7. COMMENT

In addition to all these, the results obtained are not at competitive with the results in the literature.

RESPONSE

1) We have list the values for the parameters of the experiments in a separate table.

2) We have added comparison against two relevant methods from the literarature: MRMR and PCA. The results are listed in two additional tables in the revised manuscript.

3) We have expanded the description for every method used in the experiments

Reviewer 2 Report

The presented paper briefly describes the software tool for feature construction. The author designed an exciting and useful tool based on Grammatical Evolution. The author described the problems where data classification can be used as well as basic information on used algorithms and software libraries and frameworks. There is almost no information about related or similar software, so the reader can't tell if the proposed solution is the only one. 

Description of used algorithms, methods, and features is less than minimal and has to be elaborated. This part looks like a report, no scientific paper.

Experiments are described quite precisely, but the paper lacks a discussion on the results.

Mentioned parts must be elaborated to accept the presented paper, as the proposed solution is an interesting tool to present to data analysts.

Author Response

1. COMMENT

There is almost no information about related or similar software, so the reader can't tell if the proposed solution is the only one.

RESPONSE

We have added the following paragraph in the Introduction section of the revised manuscript:

Related freely available software packages on feature selection and construction are also the Mlpack package[mlpack], that implements the PCA method, the GSL software package obtained from https://www.gnu.org/software/gsl/doc/html/index.html that also implements the PCA method among others, the MRMR package writen in ANSI C++ by Hanchuan Peng [mrmr1, mrmr2] etc.

2. COMMENT

Description of used algorithms, methods, and features is less than minimal and has to be elaborated. This part looks like a report, no scientific paper.

RESPONSE

1) We have list the values for the parameters of the experiments in a separate table.

2) We have added comparison against two relevant methods from the literarature: MRMR and PCA. The results are listed in two additional tables in the revised manuscript.

3) We have expanded the description for every method used in the experiments

3. COMMENT

Experiments are described quite precisely, but the paper lacks a discussion on the results.

RESPONSE

We have added the following text in the revised version of the manuscript in subsection 4.2

Summarizing the conclusions of the experiments, one can say that the proposed method obviously outperforms the other techniques in most cases, especially in the case of regression datasets. In the case of data classification, there is a gain of the order of 30% and in the case of regression data the gain from the application of the proposed technique exceeds 50%. The gain in many cases from the application of the proposed technique can even reach 90%. In addition, the MRMR method seems to be superior in most cases to the PCA, possibly pointing the way for a future research on the combination of Grammatical Evolution and MRMR. Furthermore, among the three cases of models used to evaluate the constructed features (FCRBF, FCMLP, FCNNC), there does not seem to be any clear superiority of any of the three. However, we would say that the nnc method slightly outperforms the simple genetic algorithm.”

Round 2

Reviewer 1 Report

The paper can be accepted in this form.